# `MACD`: Multilingual Abusive Comment Detection at Scale for Indic Languages

Vikram Gupta[1], Sumegh Roychowdhury[2],[*] Mithun Das[2], Somnath Banerjee[2], Punyajoy Saha[2], Binny Mathew[2], Hastagiri Vanchinathan[1], Animesh Mukherjee[2]

[1]ShareChat, India, [2]Indian Institute of Technology, Kharagpur, India
{vikramgupta,hasta}@sharechat.co
{sumegh01,mithundas,punyajoys,binnymathew}@iitkgp.ac.in
{som.iitkgpcse@kgpian.iitkgp.ac.in},{animesh@cse.iitkgp.ac.in}

## Abstract

Social media platforms were conceived to act as online 'town squares' where people could get together, share information and communicate with each other peacefully. However, harmful content borne out of bad actors are constantly plaguing these platforms slowly converting them into 'mosh pits' where the bad actors take the liberty to extensively abuse various marginalised groups. Accurate and timely detection of abusive content on social media platforms is therefore very important for facilitating safe interactions between users. However, due to the small *scale* and sparse *linguistic* coverage of Indic abusive speech datasets, development of such algorithms for Indic social media users (one-sixth of global population) is severely impeded. To facilitate and encourage research in this important direction, we contribute for the first time `MACD` - a large-scale (150K), human-annotated, multilingual (5 languages), balanced (49% abusive content) and diverse (70K users) abuse detection dataset of user comments, sourced from a popular social media platform - *ShareChat*[2]. We also release `AbuseXLMR`, an abusive content detection model pretrained on large number of social media comments in 15+ Indic languages which outperforms `XLM-R` and `MuRIL` on multiple Indic datasets. Along with the annotations, we also release the mapping between comment, post and user id's to facilitate modelling the relationship between them. We share competitive monolingual, cross-lingual and few-shot baselines so that `MACD` can be used as a dataset benchmark for future research. Dataset, code and `AbuseXLMR` are available at: `https://github.com/ShareChatAI/MACD`

## 1 Introduction

Adoption of social media platforms has increased dramatically in recent times. Unfortunately, this rapid adoption is often accompanied with an increase in the frequency of abusive interactions like cyber-bullying, abusive language, hate speeches etc. [67] towards individuals and groups which can trigger violent real-world situations [13], and result in devaluation and exclusion of minority members [36, 51]. Repeated exposure to these types of harmful content could lead to psychological trauma, radicalization and even self-harm [69]. In addition, several incidents in India, such as smearing campaigns against famous political leaders, celebrities, and social media personalities,

---

[*]Work done during internship at ShareChat, India.
[2]https://sharechat.com/

| Comments (Translation) | Reason | Comments (Translation) | Reason | Comments (Translation) | Reason |
|---|---|---|---|---|---|
| बुर देदो तो मूह में लंड ले लो तो
(Give pu**y and take my d**k) | Sexual | लोग कैसे कांग्रेस के 70 साल के बलिदान को भूल सकते है
(People can't forget Congress party's sacrifices in the last 70 years) | Political | सुपर वेरी गुड काला चश्मा
(Very good black shades) | Compliment |
| भोसड़ी को बात करने दो
(A**hole let me talk) | Profane | दूध का दूध पानी का पानी 😊 😊
(Sifting of just from unjust) | Quote | पारंपरिक कपडे मध्ये अप्रतिम सौन्दर्य 👌
(Amazing beauty in traditional clothes) | Compliment |
| साले कुते तू कॉंग्रेसी चमचा है
(Bl**dy dog is Congress party slave) | Political | बाप के कपड़े उतर गए बेटी के पहनवे में बेटी ने कपड़े उतर दिए फ्लोवर बढ़ाने में
(She took off her clothes to increase followers) | Implicit Shame | मोदी जी अकेले ही बोले थे और आज उनके सात करोड़ों बोल रहे है
(Modi Ji spoke alone and today his 7cr followers are speaking) | Political |
| छक्का बनेगा क्या
(Wanna be an eunuch) | Gender | रक्त नहीं वो पानी हैं, जो श्रीराम ना बोले, वो पकिस्तानी हैं। Jay Shri Ram 🙏▶
(Not chanting "Jay Shri Ram" go to Pakistan) | Religion | क्या सेक्सी है
(Very sexy) | Ambiguous |

Figure 1: Examples from MACD. *Abusive* examples along with their translations and descriptions are boxed in red color and *Non-abusive* in green. [Best viewed in color]

online anti-religious propaganda and cyber harassment [3] observed on social media platforms further encourages one to tackle this alarming problem. To counter, social media platforms employ human content moderators to filter such content, so that end-users are not exposed to these. However, content moderators have been reported to suffer from burnout, depression, and PTSD as a result of viewing these harmful content day in and day out [6]. Thus, an effective solution to combat online abuse would be to develop automatic abuse detection systems which can identify abusive content in a timely manner and could at least partially alleviate the burden from the moderators and facilitate safe and healthy interactions on social media platforms.

However, social media interactions are not structured formally and contain *spelling mistakes, slangs, grammatical errors, emoticons* etc. Moreover, the content is expressed in multiple languages and can even be code-mixed, which makes detection extremely challenging, especially for resource impoverished languages. To solve this, abusive speech detection datasets have been developed for various languages [66, 28, 73, 40, 22, 24, 72, 29, 7, 54, 61, 11, 41, 5, 60, 12, 47, 27, 50, 52, 8, 55, 34] and have been primarily sourced from social media platforms like Twitter, Facebook, Gab, YouTube etc. Similarly, abusive speech datasets for Indic languages have also been contributed [48, 47, 62, 70, 9]. However, the *scale* and *linguistic coverage* of these datasets is sparse (see Table 1). Large-scale annotated datasets for abusive speech research in Indic languages is a need of the hour. Considering that one-sixth of the global population speaks Indic languages, such dataset would have a huge impact. While combining smaller datasets [58] into one large-scale dataset is possible, differences in annotation guidelines and dataset sources (Gab, YouTube, Reddit etc.) can introduce *inconsistency*, which can impact the studies. In such scenarios, a large-scale dataset curated from a single source and annotated based on a consistent set of guidelines is more helpful.

To reduce this gap and foster abusive speech detection in Indic languages, we contribute a novel, large-scale, human-annotated, well-balanced, diverse and multilingual abuse detection dataset - **M**ultilingual **A**busive **C**omment **D**etection dataset (MACD), sourced from a popular social media platform - *ShareChat*, which supports over 15 Indic languages. MACD comprises of 150K textual comments posted on 92881 posts by 70453 users with 74K *abusive* and 77K *non-abusive* comments (49% abuse ratio) from five Indic languages - *Hindi (Hi), Tamil (Ta), Telugu (Te), Malayalam (Ml) and Kannada (Kn)*. We select these languages as they witness maximum engagement on the platform. To the best of our knowledge, MACD is one of the largest abusive speech datasets for Indic languages. Comments containing abusive speech towards individuals/religions/race/political group, sexual references, profane language, violent intentions etc. are annotated as *abusive* by language-specific team of expert annotators.

Along with MACD, we further contribute AbuseXLMR, which has been pretrained using the XLM-R [19] model from 5M+ social media comments. AbuseXLMR bridges the domain gap which exists in XLM-R, mBERT and MuRIL as they have not been pretrained over social media datasets. We show that AbuseXLMR excels over these contextual models on MACD as well as many other popular Indic abuse detection datasets like HASOC [48], MOLD [30], and Bengali [59] datasets. AbuseXLMR also triumphs over XLM-R and MuRIL under zero-shot cross-lingual and few-shot performance settings

---

[3]https://tinyurl.com/3fa5zdvy, https://tinyurl.com/mryj7jkd, https://tinyurl.com/4pcs7frf, https://tinyurl.com/8b2c35u3

Table 1: Comparison of `MACD` with **Indic** abusive speech datasets (>4K samples)

| Dataset | Language | Samples | Abuse% |
|---------|----------|---------|--------|
| | Indic | | |
| [48] | Hindi | 6K | 51% |
| [47] | Hindi | 5K | 50% |
| [42] | Hindi | 4.5K | 32% |
| [70] | Hindi | 4.5K | 32% |
| [59] | Bengali | 30K | 33% |
| | MACD | | |
| MACD | Hindi | 33K | 52% |
| MACD | Tamil | 30K | 46% |
| MACD | Telugu | 30K | 52% |
| MACD | Kannada | 33K | 49% |
| MACD | Malayalam | 25K | 45% |

Table 2: `MACD` statistics.

| Data description | Value |
|------------------|-------|
| # Total samples | 152422 |
| # Abusive samples | 74550 |
| # Non-abusive samples | 77872 |
| # Abuse % | 49% |
| # Posts | 92881 |
| # Users | 70453 |
| # Average comments length | 85 chars |
| # Shortest comment | 2 chars |
| # Longest comment | 6621 chars |

highlighting the improved low-data capabilities. Thus `AbuseXLMR` positions itself as a domain-adapted, data-efficient and accurate abuse detection model for Indic languages.

The *scale*, *linguistic coverage* and *consistent* expert level annotations of `MACD` for resource-impoverished Indic languages would enable detailed study of abusive speech in these under-explored languages. Owing to the large *scale* of `MACD` could facilitate both pretraining and end-to-end training of deep models as we show using a series of competitive baselines here.

- We release `MACD`, a large-scale (150K), well-balanced (49% abuse ratio), human-annotated, multilingual (5 Indic languages) and diverse (70K users) abuse detection dataset.
- We release `AbuseXLMR` which is a pretrained abuse detection model for social media content in Indic languages. `AbuseXLMR` outperforms `XLM-R` and `MuRIL` on four Indic datasets which can be used for future research.
- We contribute monolingual, cross-lingual and few-shot baselines for benchmarking and future work on our dataset.

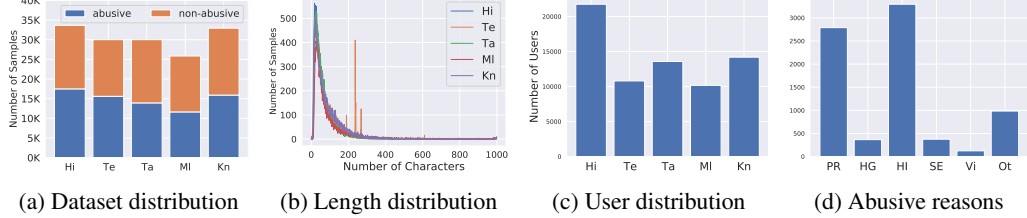

(a) Dataset distribution    (b) Length distribution    (c) User distribution    (d) Abusive reasons

Figure 2: `MACD` across multiple dimensions: (a) number of samples for *abusive* and *non-abusive* categories for all the languages (b) distribution of length (number of characters) of comments in each language (c) distribution of number of users for each language (d) distribution of sub-categories on a subset of abusive comments from Hindi language. [Best viewed in color].

## 2   Related work

Abusive speech detection has received lot of attention from the research community across textual [25, 23, 28, 22], audio [37, 33] and visual [35, 31, 4] domains. Abuse detection datasets in non-Indic languages [34, 28, 73, 52, 40, 22, 24, 72, 47, 29, 7, 52, 54, 41, 50, 52, 3, 2, 8, 55] have been instrumental in pushing the state-of-the-art for these languages. We summarize abuse detection text datasets in the non-Indic language subsection of Appendix Table 10. In contrast to non-Indic datasets, we observe that Indic datasets are substantially small in scale. [48, 47] proposed dataset for Hindi language consisting of 5K and 6K samples respectively. While the dataset is well balanced with 50% abusive content, the number of samples are insufficient for large-scale study. [62] proposed a dataset for Hindi language containing 2K posts sourced from Twitter and Facebook. Similarly, [70] contribute a dataset for Hindi (4.5K posts) and Marathi (2K posts). [9] proposed a dataset for

Bengali language containing 4K posts. [59] contribute a large (30K) dataset for Bengali but the abuse ratio is lower (33%). In Table 1, we note the statistics of Indic datasets containing more than 4K samples and observe that majority of these datasets are relatively small. We also compare MACD with code-mixed datasets in Appendix Table 13. The code-mixed dataset repository[4] is one of the largest of its kind comprising more than 10 Indic languages and 740K comments. However, it is not available publicly for research purposes. Unlike code-mixed datasets, MACD is focussed towards studying Indic languages and the vocabulary contains less than 4% code-mixing (see Table 12). Overall, MACD is large-scale, publicly available and encompasses five popular Indic languages - *Hindi, Tamil, Telugu, Kannada* and *Malyalam* with consistent annotation for Indic languages.

**Existing methods**: Initial investigations in abusive speech detection leveraged lexicon [63], hand-crafted features [22, 71, 41] and metadata [68, 18, 57]. Recently, transformer based models have shown state-of-the-art performance for various hate speech detection [46, 64, 14, 49]. Multilingual variants like mBERT [26], XLM-R [20] have been proposed for addressing semantic understanding across multilingual and resource-impoverished settings. For Indic languages, MuRIL [44] and IndicBERT [43] have been proposed. MuRIL has been trained over 16 Indic languages and English language datasets using MLM [65] and translation language modelling (TLM) [21]. IndicBERT is a multilingual ALBERT [45] model trained over 12 Indian languages. For all these models, the pretraining is not done social media data; in order to bride this gap we release AbuseXLMR, our own pretrained model on 5M+ social media data. Besides a limited number of studies have been explored on the effect of transfer learning [56]. Therefore we explore this gap by studying various transfer schemes, i.e., zero-shot learning and few-shot learning.

Table 3: **Monolingual:** Accuracy (Acc) and F1-macro score (F1) for different models on MACD dataset. Best results and second best results are shown in **bold** and underline respectively.

| Model | Hindi | | Tamil | | Telugu | | Kannada | | Malayalam | |
|---|---|---|---|---|---|---|---|---|---|---|
| | Acc | F1 | Acc | F1 | Acc | F1 | Acc | F1 | Acc | F1 |
| TF-IDF (LR) | 81.23 | 81.19 | 83.57 | 83.41 | 86.10 | 86.08 | 82.12 | 81.94 | 81.84 | 81.52 |
| TF-IDF (SVM) | 82.36 | 82.34 | 84.43 | 84.33 | 86.52 | 86.49 | 83.13 | 83.04 | 83.66 | 83.42 |
| mBERT [26] | 84.32 | 84.31 | 87.42 | 87.37 | 89.08 | 89.07 | 86.64 | 86.58 | 84.33 | 84.18 |
| XLM-R [20] | 86.12 | 86.11 | 87.92 | 87.87 | 89.50 | 89.44 | 86.75 | 86.71 | 85.55 | 85.42 |
| MuRIL [44] | 85.72 | 85.68 | 88.35 | 88.33 | 89.47 | 89.42 | 87.20 | 87.12 | 85.49 | 85.32 |
| AbuseXLMR | **87.96** | **87.93** | **88.62** | **88.60** | **91.40** | **91.37** | **88.14** | **88.12** | **88.14** | **88.04** |

Table 4: **Alternate Splits:** Accuracy (Acc) and F1-macro score (F1) for different splits using AbuseXLMR.

| Splits | Hindi | | Tamil | | Telugu | | Kannada | | Malayalam | |
|---|---|---|---|---|---|---|---|---|---|---|
| | Acc | F1 | Acc | F1 | Acc | F1 | Acc | F1 | Acc | F1 |
| Random (80:10:10) | 88.53 | 88.52 | 88.83 | 88.81 | 91.63 | 91.60 | 88.00 | 88.00 | 88.23 | 88.14 |
| Chronological | 87.43 | 86.78 | 87.47 | 86.99 | 90.42 | 90.38 | 87.16 | 87.16 | 83.17 | 81.82 |
| Unbalanced | 93.05 | 85.69 | 92.19 | 84.48 | 94.65 | 88.86 | 93.33 | 85.74 | 92.27 | 82.81 |

## 3 MACD **dataset**

### 3.1 **Dataset collection**

Comments have been sourced from a popular social media platform - *ShareChat*. Since *abusive* comments are rare, we sample textual comments which have been reported as abusive by other users on the platform. These comments have higher probability of being abusive. We further enhance this set by using keyword matching (lexicon of 15K trigger words) for identifying comments containing frequently used abusive words. However, due to the contextual nature of abuse, presence of these words is not sufficient and manual annotation is required for assigning the ground-truth. In order to obtain the language of the comment, we used the language specified by the user in her profile

---

[4]https://www.kaggle.com/competitions/iiitd-abuse-detection-challenge/overview

Table 5: **Zero-shot cross-lingual**: Accuracy (Acc) and F1-macro score (F1) in zero-shot cross-lingual setting. The models are trained on the *source* language (row) and evaluated on the test set of *target* languages (column). Best cross-lingual results are marked in **bold** and monolingual results are highlighted by underline.

| Target → | Hindi | | Tamil | | Telugu | | Kannada | | Malayalam | |
|---|---|---|---|---|---|---|---|---|---|---|
| Source ↓ | Acc | F1 | Acc | F1 | Acc | F1 | Acc | F1 | Acc | F1 |
| MuRIL | | | | | | | | | | |
| Hindi | 85.72 | 85.68 | 75.35 | 75.12 | **70.30** | **70.19** | 72.82 | 72.79 | 67.91 | 67.16 |
| Tamil | 70.20 | 68.84 | 88.35 | 88.33 | 69.55 | 68.61 | 68.36 | 66.34 | 68.51 | 66.74 |
| Telugu | **75.99** | **75.85** | 78.67 | 78.50 | 89.47 | 89.42 | **73.33** | **72.90** | 72.88 | 71.51 |
| Kannada | 63.33 | 59.68 | 66.81 | 62.12 | 56.72 | 50.84 | 87.20 | 87.12 | 63.25 | 57.47 |
| Malayalam | 68.25 | 66.63 | 73.62 | 72.43 | 66.17 | 64.74 | 68.53 | 67.59 | 85.49 | 85.32 |
| XLM-R | | | | | | | | | | |
| Hindi | 86.12 | 86.11 | **72.35** | **71.82** | **70.28** | **70.16** | **71.07** | **71.07** | **74.45** | **74.17** |
| Tamil | 58.78 | 53.59 | 87.92 | 87.87 | 54.62 | 47.57 | 57.57 | 48.40 | 60.50 | 49.18 |
| Telugu | 61.59 | 58.30 | 67.83 | 64.83 | 89.50 | 89.44 | 68.47 | 67.11 | 66.67 | 62.08 |
| Kannada | 60.97 | 57.02 | 60.90 | 53.29 | 56.60 | 50.84 | 86.75 | 86.71 | 62.92 | 55.21 |
| Malayalam | **67.20** | **65.98** | 69.78 | 68.26 | 66.35 | 65.41 | 66.92 | 65.97 | 85.55 | 85.42 |
| AbuseXLMR | | | | | | | | | | |
| Hindi | 87.96 | 87.93 | 86.73 | 86.70 | 81.50 | 81.49 | 82.05 | 82.01 | **84.60** | **84.15** |
| Tamil | 81.21 | 81.02 | 88.62 | 88.60 | 79.83 | 79.74 | **84.24** | **84.15** | 80.77 | 79.51 |
| Telugu | **85.75** | **85.74** | **87.08** | **87.05** | 91.40 | 91.37 | 83.54 | 83.54 | 84.35 | 83.93 |
| Kannada | 69.93 | 67.87 | 83.13 | 82.55 | 77.45 | 77.01 | 88.14 | 88.12 | 74.87 | 71.57 |
| Malayalam | 84.78 | 84.78 | 85.95 | 85.94 | **82.48** | **82.48** | 82.75 | 82.72 | 88.14 | 88.04 |

to collect language specific comments. However, we found that majority of users switch between different languages, so user-specified language is not reliable. To address this, we use linguistic rules[5] and human-annotation for labelling the comment with the correct language tag. We also ensured that MACD has less code-mixing to focus primarily on Indic languages and vocabulary by removing comments containing higher proportion of Roman characters. As shown in Appendix Table 12, MACD has less than 4% code-mixing. This data was collected for a period of six months (Sep 2021 - Feb 2022). To mitigate the presence of user bias in the dataset, we threshold the number of comments fetched for each user by 500 while preparing the dataset. Personally Identifiable Information (PII) like names, phone numbers, email addresses, social media handles etc. present in the posts were removed to protect user privacy. The identifier of posts, comments and users were randomized for maintaining anonymity. Emojis were preserved for capturing social media nuances. More details about preprocessing in Appendix A.2.

### 3.2 Annotation

*Annotation team*: For each language, two native speakers of that language are selected as annotators and were employed on a contract basis for annotating the dataset. Overall, MACD has been annotated by 53 expert-trained annotators out of which 7 identified themselves as *female*, while remaining as *male*. Majority of the annotators had age between 20 to 27 years with 25 as the average age. These annotators are expert in annotating and moderating social media content and work full-time on related assignments. On average, five comments per minute have been annotated by each annotator.

*Annotation labels*: All the comments in MACD are manually annotated by the annotation team with binary labels (*abusive* and *non-abusive*) representing the presence/absence of abuse. In the case of disagreements, the final label was selected by a senior annotator, chosen on the basis of years of experience in annotating social media content. After annotation, we sampled comments across categories and languages for creating a well-balanced and diverse dataset.

*Annotation guidelines*: Considering the complexity and nuances of abuse identification, following guidelines were shared to ensure consistent annotations. Comments expressing the following intentions should be annotated as *abusive*. (a) *Profanity* (PR): Comments containing profane, cuss, swear words are annotated as *abusive*, (b) *Sexual references* (SE): Comments which contains sexual

---

[5]We use character set of these languages to identify the dominant language of the comment.

Table 6: **Few-shot**: Accuracy (Acc) and F1-macro (F1) under few-shot (5, 25, 100, 250 and 500 samples) settings. We run experiments for five seeds and report the average performance.

| Language | 5 | | 25 | | 100 | | 250 | | 500 | |
|---|---|---|---|---|---|---|---|---|---|---|
| | Acc | F1 | Acc | F1 | Acc | F1 | Acc | F1 | Acc | F1 |
| XLM-R | | | | | | | | | | |
| Hindi | 50.36 | 36.79 | 51.97 | 39.70 | 55.78 | 48.49 | 69.57 | 69.22 | 75.88 | 75.67 |
| Tamil | 49.45 | 33.33 | 52.06 | 43.84 | 58.18 | 49.93 | 72.18 | 71.54 | 78.26 | 78.22 |
| Telugu | 51.98 | 36.71 | 52.16 | 38.97 | 53.92 | 40.07 | 77.09 | 77.01 | 80.32 | 80.23 |
| Kannada | 50.20 | 35.56 | 49.58 | 33.13 | 54.91 | 47.94 | 70.51 | 70.41 | 75.41 | 75.29 |
| Malayalam | 46.91 | 32.52 | 52.31 | 46.02 | 55.25 | 45.94 | 70.68 | 70.45 | 77.21 | 77.06 |
| MuRIL | | | | | | | | | | |
| Hindi | 46.95 | 34.41 | 50.32 | 35.18 | 55.71 | 44.18 | 60.99 | 50.29 | 78.17 | 78.14 |
| Tamil | 51.46 | 36.49 | 52.05 | 34.21 | 54.38 | 40.01 | 71.57 | 67.79 | 79.52 | 79.48 |
| Telugu | 48.06 | 32.44 | 49.36 | 33.70 | 55.98 | 44.35 | 77.37 | 77.26 | 80.26 | 80.24 |
| Kannada | 51.18 | 36.87 | 51.88 | 36.53 | 57.13 | 48.76 | 62.66 | 55.66 | 77.15 | 77.06 |
| Malayalam | 52.21 | 36.77 | 53.22 | 35.07 | 55.97 | 41.70 | 67.69 | 63.40 | 74.75 | 74.45 |
| AbuseXLMR | | | | | | | | | | |
| Hindi | 52.40 | 49.25 | 60.22 | 56.22 | 77.31 | 77.05 | 82.29 | 82.25 | 84.56 | 84.51 |
| Tamil | 53.47 | 48.84 | 61.80 | 54.80 | 81.12 | 81.01 | 84.25 | 84.20 | 86.13 | 86.09 |
| Telugu | 52.63 | 42.15 | 55.01 | 45.45 | 82.96 | 82.77 | 87.65 | 87.63 | 88.77 | 88.71 |
| Kannada | 52.17 | 43.20 | 56.85 | 47.22 | 80.43 | 80.28 | 84.35 | 84.29 | 85.32 | 85.23 |
| Malayalam | 48.07 | 42.29 | 57.33 | 50.41 | 78.24 | 78.01 | 83.15 | 82.97 | 84.68 | 84.52 |

references, (c) *Personal beliefs and practices* (HI): Comments in which the dressing sense, choice of content, choice of language etc. are targeted, (d) *Gender discrimination* (HG): Comments in which the person is attacked on basis of gender, (e) *Religious beliefs and practices* (HR): Comments in which the person is attacked on basis of religious beliefs and practices. For example, comments questioning wearing of head-scarf, (f) *Hate towards political views* (HP): Comments in which the political views of person are attacked. For example, ridiculing people for supporting political party, (g) *Violent intent* (VI): Comments in which threat or call for violence is raised. In Figure 1, we share examples from Hindi language.

*Annotator agreement*: We measure the annotator agreement for all the languages using Cohen's Kappa and observe $\kappa = 0.73$, $\kappa = 0.72$, $\kappa = 0.71$, $\kappa = 0.69$, and $\kappa = 0.70$ for Hindi, Tamil, Telugu, Malayalam and Kannada respectively.

*Metadata:* We also include the identifier of user who made the comment and identifier of the original content on which the comment was expressed to further enrich MACD dataset with social graph information. We masked both these identifiers for respecting privacy of the users.

### 3.3 Dataset analysis

We summarize the key statistics of MACD in Table 2.

**Linguistically diverse**: MACD has been sourced from five Indic languages - Hindi, Tamil, Telugu, Malayalam and Kannada providing a highly diverse multilingual abuse detection dataset.

**Large-scale**: MACD dataset contains 150K samples with more than 25K samples for each languages which makes it one of the largest abuse detection dataset for Indic languages.

**Balanced**: MACD is balanced across both categories with 74K *abusive* and 77K *non-abusive* comments (49% abusive samples). Ratio of abusive comments range from 52%, 46%, 52%, 48% and 49% for Hindi, Tamil, Telugu, Malayalam and Kannada respectively as shown in Figure 2(a).

**Comment length**: We plot the distribution of number of characters present in the comments for all the languages in Figure 2 (b). We note that majority of comments have an average of 85 characters per

comment which reflects the spontaneous and conversational nature of these comments. The shortest comment in our dataset is 2 characters and longest is 6621 characters[6].

**User distribution**: In Figure 2 (c), we plot the distribution of users across languages. We note that all the languages have more than 10K different users with Hindi having more than 20K users. This shows that MACD is rich in diversity because it captures wide range of nuances like spelling and grammatical mistakes, use of abbreviations and emojis along with various social sensitivities and beliefs of large number of social media users.

**Abuse types**: We also investigate the diversity of MACD by having some crude annotations of the different abuse types for a small subset of Hindi abusive comments. As observed in Figure 2 (d), HI forms the dominant subcategory where comments are being made toward personal beliefs and practices. This is followed by profanity (PR) where comments containing *cuss, swear* words are being made. We would like to point out that this is an ongoing work and we are in the process of refining the type labels and extending it to the full dataset. In a future work, we shall release this fine-grained data once it has been satisfactorily tested for quality.

**Dataset splits**: We provide different MACD splits for analyzing various aspects of MACD:

(a) Random splits: We randomly split MACD in 60:20:20 ratio to form the train, validation and test set and use this as default split for most of the study. We also release 80:10:10 ratio splits of MACD to increase the amount of training data for improved performance.

(b) Chronological splits: Abusive behaviour evolves over time inspired by real-world events. For modelling these trends, we split MACD, chronologically into 60:20:20 ratio. We ordered all the comments using their date of creation before splitting them.

(c) Unbalanced splits: Abusive content is rarely balanced and is rather sparse in natural settings. To simulate this scenario, we also provide an unbalanced split for MACD where we sample *abusive* and *non-abusive* comments in 1:5 ratio to represent near-natural settings.

## 4  AbuseXLMR

Contextual models are trained on large-scale multilingual datasets but are not adapted for social media domain. This creates a domain gap and results in inferior performance, especially in low-data scenarios. To tackle this, we pretrain XLM-R on large-scale dataset extracted from *ShareChat*. We select XLM-R for pretraining because it showed good performance (Table 3). Moreover, unlike MuRIL and mBERT, XLM-R does not require consecutive sentences corpora because it does not use Next Sentence Prediction (NSP) task during pretraining. Since, MACD comments are primarily single-sentence and consecutive comments are not coherent, it is not useful to formulate MACD for NSP task. We extract large amounts of unlabelled comments which have been reported as *abusive* by *ShareChat* users or matched one of the trigger word for a duration of one year (Apr 2021 - Apr 2022) from the platform. We randomly sample 5M comments out of the complete corpora and use these sampled comments for continued pretraining of the XLM-R model using masked language modelling (MLM) loss. These comments belonged to 15+ Indic languages extending to Bengali, Marathi, etc. which makes the pretraining corpus linguistically diverse and qualifies AbuseXLMR as a suitable model for multiple Indic languages. Language set by the user was used as proxy since accurate determination of language is not required as we are pretraining with the complete corpus. Pretraining on this corpus adapts AbuseXLMR to the social media nuances like spelling mistakes, grammatical mistakes, emoticons etc. and thus enhances its capabilities as compared to MuRIL and XLM-R. Our experiments demonstrate the efficacy of AbuseXLMR over other models highlighting the importance of bridging the domain-gap across MACD and other popular Indic datasets.

## 5  Experiment and results

We consider the task of classifying textual comment into *abusive* and *non-abusive* categories. Unless specified, we use the random 60:20:20 splits for our experiments. We compute the performance of TF-IDF based model and transformer-based multilingual contextual models like XLM-R [20],

---

[6]We report characters instead of words because due to the informal nature of social media content, lot of comments did not use standard delimiters like *space*

Table 7: Accuracy and macro-F1 in monolingual (*mono*), joint (*joint*) and pretraining (*pr-mono*) settings.

| Model | Hindi | | Tamil | | Telugu | | Kannada | | Malayalam | |
|---|---|---|---|---|---|---|---|---|---|---|
| | Acc | F1 | Acc | F1 | Acc | F1 | Acc | F1 | Acc | F1 |
| mBERT (*mono*) | 84.32 | 84.31 | 87.42 | 87.37 | 89.08 | 89.07 | 86.64 | 86.58 | 84.33 | 84.18 |
| mBERT (*joint*) | 85.46 | 85.44 | 87.92 | 87.89 | 90.05 | 90.00 | 87.16 | 87.13 | 85.28 | 85.17 |
| XLM-R (*mono*) | 86.12 | 86.11 | 87.92 | 87.87 | 89.50 | 89.44 | 86.75 | 86.71 | 85.55 | 85.42 |
| XLM-R (*joint*) | 85.27 | 85.26 | **88.15** | **88.10** | 89.08 | 89.74 | **87.48** | **87.42** | 86.09 | 85.87 |
| XLM-R (*pr-mono*) | **86.34** | **86.32** | 87.90 | 87.88 | **90.05** | **90.00** | 87.20 | 87.17 | **86.54** | **86.47** |

Table 8: **Cross dataset**: Accuracy and macro-F1 on HASOC-2019 using MACD with XLM-R.

| Model | Acc | F1 |
|---|---|---|
| Baseline | 83.08 | 82.82 |
| Pretrained-MACD | 83.68 | 83.99 |
| Hindi-MACD | 83.58 | 83.76 |
| Joint-MACD | **84.37** | **84.26** |
| Zero-Shot-MACD | 76.56 | 76.39 |

Table 9: **Cross dataset**: Accuracy and macro-F1 score on MOLD [30], Bengali [59] and HASOC [48] dataset.

| Model | Marathi [30] | | Bengali [59] | | Hindi [48] | |
|---|---|---|---|---|---|---|
| | Acc | F1 | Acc | F1 | Acc | F1 |
| MuRIL [44] | 89.60 | 88.38 | 90.63 | 89.59 | 84.45 | 84.22 |
| XLM-R [20] | 88.96 | 87.70 | 90.47 | 89.32 | 82.85 | 82.39 |
| AbuseXLMR | **90.72** | **89.50** | **90.78** | **89.73** | **84.98** | **84.75** |

mBERT [26], MuRIL [44] and AbuseXLMR under different settings. All the five languages included in MACD are covered in the training corpus of the contextual models making them suitable for our study. We discuss more details about the models and training in the appendix sectionA.5. We report accuracy and macro-F1 score on the test split of MACD for measuring the model's performance.

**Monolingual experiments**: We show the performance of monolingual experiments (training and testing on the same language) in Table 3. We observe that contextual models improve upon the TF-IDF results by nearly 4%. This shows the advantages of modeling context in our dataset. The superior performance of contextual models can also be attributed to the fact that all the five languages of MACD were also used in the pretraining stages of these models. Comparing the contextual models, MuRIL performs better than XLM-R and mBERT for Tamil and Kannada, while XLM-R shows best performance for Hindi, Telugu and Malayalam. Improved performance of XLM-R can be attributed to the fact that it was pretrained on more than 100 languages including the Indian languages while MuRIL was trained specifically on 17 Indic languages. Finally, we note that AbuseXLMR, outperforms all the other models for all the languages highlighting the importance of pretraining on domain-aligned data.

**Other splits**: We further perform monolingual experiments with AbuseXLMR in Table 4 for the other three splits of MACD. We observe that the F1 scores drop, while accuracy improves on *unbalanced splits*. This could be due to the high class imbalance. We also note that the model performance drops when we use chronological splits highlighting the impact of evolving trends. In future, we would like to investigate how additional signals from the social network could be used to push up the performance back for both the above scenarios.

**Zero-shot cross-lingual experiments**: We compute the performance for each language in a zero-shot cross-lingual setting. We train the model on the *source* language and measure the performance against the test set of the *target* languages. From Table 5, we observe cross lingual performance is substantially lower (more than 10% drop in F1) than monolingual performance for XLM-R and mBERT model. This shows that both these models do not generalize so well across languages. However, the cross-lingual performance of AbuseXLMR is substantially higher than both XLM-R and MuRIL. These results show that domain-adaption of AbuseXLMR improves the zero shot performance drastically.

**Few-shot experiments**: In Table 6 we finetune models on 5, 25, 100, 250 and 500 samples for five random seeds and report the average performance. We observe that using less than 100 samples with XLM-R and MuRIL results in macro-F1 score of less than 50%, highlighting that 100 and fewer examples are not sufficient. However, with 250 and more examples, the performance starts to improve. We note that AbuseXLMR demonstrates improved few-shot capabilities. Even with 100

samples, `AbuseXLMR` is able to demonstrate impressive results, almost comparable to performance shown by `MuRIL` and `XLM-R` with 500 samples.

**Joint training experiments**: In Table 7, we study the impact of using all the languages present in `MACD` together for training the models to investigate if cross-learning between languages happens. We combine the training splits of all the languages to form a combined training set. We repeat the same for the validation set. We then train the model using these combined sets and test them on the same held out test set of the respective languages. We note that joint training (*joint*) using `mBERT` improves the performance for all the languages over the monolingual (*mono*) setting, while `XLM-R` improves the results for three languages. These results show that there is cross-learning between the languages, which can be leveraged. Moreover, since social media content contains code-switching and emojis, there could be an overlap of tokens between languages that further benefits the *joint* training.

**Pretraining experiments**: We study the impact of self-supervised pretraining using training set of all the languages using Masked Language Modelling (MLM) loss. We continue pretraining of the `XLM-R`[7] checkpoint for 10 more epochs using the training data for all the 5 languages of `MACD`. We then finetune this checkpoint with monolingual supervised dataset for each language. We observe that this two-stage process of pretraining followed by finetuning shows performance gains for all the languages over their monolingual (*mono*) settings as shown by the *pr-mono* row in Table 7. These highlight the efficacy of using `MACD` for pretraining models for other datasets also.

**Cross-dataset experiments**: We evaluate the performance of models trained using `MACD` on HASOC-2019 binary classification[8] task using the Hindi subset. Since the samples oh HASOC are collected from Twitter and Facebook, this also allows to understand the *cross-platform* learning since `MACD` has not been sourced from Twitter/Facebook. We finetune `XLM-R` under different settings and evaluate performance on the HASOC test set.(a) **Baseline:** Finetune `XLM-R` on HASOC train set (b) **Pretrained-`MACD`:** We pretrain `XLM-R` over `MACD` dataset and finetune this checkpoint on train set of HASOC (c) **Hindi-`MACD`** We finetune the `XLM-R` model in supervised manner on Hindi subset of `MACD` dataset and finetune it on HASOC train set (d) **Joint-`MACD`:** We select the `XLM-R` model trained in supervised manner on combined `MACD` dataset and finetune it on HASOC train set (e) **Zero-shot-`MACD`:** Evaluate **joint** model on HASOC test set in zero-shot setting. From Table 8, we observe that self-supervised pretraining `XLM-R` using `MACD` dataset improves F1-macro score from 82.82 to 83.99. Finetuning `XLM-R` using supervision of the Hindi set from `MACD` also improves upon the baseline results by more than **1%**. However, finetuning the `XLM-R` trained over combination of all the languages from `MACD` on the HASOC train show the best performance by improving the baseline by **1.7%** (82.82 to 84.26). This shows that combination of all the languages of `MACD` are able to improve the performance in a cross-platform setup also.

`AbuseXLMR` **Experiments:** In Table 9, we compare the performance of finetuning contextual models with three different datasets sourced from *different languages* and *platforms* to benchmark the portability and generalization of `AbuseXLMR`. We use Bengali [59] dataset, Marathi [30] dataset (MOLD) and HASOC [48] for this study. For all three datasets, we note that `AbuseXLMR` outperforms both `MuRIL` and `XLM-R`, highlighting the strength of `AbuseXLMR` in abusive content detection.

We also report results for translation based experiments A.4) to motivate the need for large-scale abusive speech datasets in Indic languages. We observe that models trained over English abuse detection datasets do not transfer well on English translated version of Indic comments.

## 6 Error analysis

`XLM-R` Errors: We analyze the failure cases of `XLM-R` since it obtains the best results for majority of the languages (Table 3). We randomly sample few error cases[9] and analyse them to understand the scenarios where contextual model fails. Based on our analysis we divide the error cases into the following categories: (a) *Implicit Abuse*: These cases do not contain explicitly abusive words and require higher order reasoning. For example, *(a)*ईद पर बकरे काटे जाते हैं तो प्रदूषण खराब नहीं होता है दीपावली पर पटाखे छोड़ते हैं तो प्रदूषण... *(Hindu festivals cause environmental harm but sacrificing animals on Eid does not?)* does not have any explicit abusive word but is aimed towards spreading hatred against a religion, (b) *Trigger Words*: While

---

[7]We perform pretraining using `XLM-R` only because `MuRIL` required paired corpora for pretraining.

[8]We experiment with Subtask A which is a binary classification between *Non-Hate* and *Hate* labels.

[9]See Appendix A.6 for more error analysis.

`XLM-R` models context well, but we still observe scenarios where it fails due to presence of concepts which dominate abusive comments. For example, a comment like बसंती इन कुत्तों के सामने नहीं नाचना😄😄😄😄😄 *(Basanti, don't dance in front of these dogs)*, is a famous movie dialogue which is non-abusive but since the word *dog* is used in abusive context, the model fails, and, (c) *Annotator Confusion*: These are ambiguous instances like *"Your figure is sexy"*, where the model prediction cannot be deemed incorrect as the comment could be interpreted in both ways depending on the cultural sensitivities.

`XLM-R` vs `TF-IDF`: We compare the error cases of `XLM-R` and `TF-IDF` to understand the impact of modelling the context for `MACD`. (a) *Contextual Abuse*: Some words appear more frequently in the abusive samples. However, depending upon the context, the meaning changes. For example, कपड़े की दुकान है *(this is cloth shop)*, the word *cloth* is used in abusive comments which question the lifestyle choices. Contextual model `XLM-R` predicts the correct label while `TF-IDF` fails, reiterating the importance of modelling the context, (b) *Spelling Mistakes*: Profane words are often misspelled, either intentionally to escape moderation algorithm or unintentionally due to informal nature of social media. For example, Hindi translation of word *a\*\*hole* (translated to English for better explanation) is spelled differently as *asshole, assh0le, a\*\*hole* etc. Token based approaches fail to capture all the variations and hence, language models pretrained with subword tokenization can detect such cases.

# 7 Broader impacts, limitations and ethical considerations

Developing large-scale, multilingual and human-annotated datasets for modeling Indic languages remains an open challenge. Absence of large-scale Indic datasets for *abuse* detection have severely impeded the research in these languages which are spoken by large number of people across the world. We hope `MACD` and `AbuseXLMR` will motivate and enable the research community to study and arrest the ill-effects of social media abuse and foster healthy, inclusive and safe social media interactions between users from all religions, genders, and ethnicity.

**Limitation**: Considering that `MACD` covers only 5 Indic languages and does not represent the entire population, more parallel efforts need to continue for further narrowing this gap.

**Explicit warning**: We request the community to be mindful that `MACD` contains comments which express abusive behaviour towards religion, region, gender etc. that might be abusive and depressing to the researchers. We did not censor such harmful words/phrases because that would defeat the purpose of the study. Kindly use your discretion while following up on our work.

**User privacy**: Protecting the privacy of users is a core value for *ShareChat* and we took measures for ensuring that no Personally Identifiable Information (PII) is made public. We will also provide an opt-out form for users to request explicit deletion of comments. We do not store the raw data used for this study. Only the anonymized data will be made available for future research.

**Informed consent**: The comments in `MACD` are publicly available on the *ShareChat* application. These comments are published by users of the platform for public consumption and informed consent is requested by the platform for broadcasting them.

# 8 Conclusion and future work

Detection of abusive content is an important problem for ensuring safe and healthy social media interactions. However, the development of automatic abusive speech detection algorithms for Indic languages is severely hampered due to the absence of large scale datasets. To reduce this gap, we contribute `MACD`, a large-scale, human annotated, well balanced, diverse and multilingual dataset posted on a popular social media platform. We also release `AbuseXLMR`, a domain-adapted model for Indic abuse detection which outperforms state-of-the-art contextual models and also demonstrates improved cross-lingual and few-shot performance. We release masked post and user identifier for constructing various social graphs like user-post, user-user and post-post to further leverage social graphs for enhancing the abuse detection performance on `MACD` dataset. In future we plan to extend `MACD` with more Indic languages and fine-grained annotations with real-valued scores [38] to enrich the annotations and further mitigate the subjectivity. We will also work toward enriching `MACD` with code-mixed comments where Indic languages are expressed using Roman characters as these scenarios are challenging and prevalent on social media platforms.

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
