# OpenReview forum: "Multilingual Abusive Comment Detection at Scale for Indic Languages"
_NeurIPS.cc/2022/Track/Datasets_and_Benchmarks — NeurIPS 2022 Datasets and Benchmarks _

### Official Review · Reviewer_SYqh · 2022-07-23
**Improved after rebuttal**

**Rating:** 5
**Confidence:** 5

**Strengths:**

The paper presents 75k social media comments annotated with labels of abusive content classes (plus 75k non-abusive comments).
This is the largest dataset for indic languages for toxic comments so far.

**Weaknesses:**

The non-abusive comments were sampled randomly. This has two problems: 1. It is unclear whether you checked these comments manually to guarantee that they are not abusive. 2. Difficult or ambiguous cases are most likely not part of this set.

Given good approaches for transfer learning exist, as well as high quality machine translation, it is unclear, whether we really need a lot of annotated data in all languages for all tasks to perform machine learning.

Half of the paper is about binary abusive language classification. The only connection to the dataset is that the test split was used for evaluation. It is unclear, what the contribution of this second half is. In particular, it is very strange that no multi-class classification setup was chosen given the multi-class labels of the presented dataset.

The AbuseXMLR method is just an XML-R model pre-trained on different data. This is not really a research contribution...



**Additional Feedback:**

There are many arxiv references. The authors should cite peer-reviewed versions instead, e.g., [18] was published at ACL.

**Clarity:**

The first half of the paper is well written, but many crucial deals are missing in the paper (they are in the supp. materials, though).

**Correctness:**

The non-abusive comments were sampled randomly. This has two problems: 1. It is unclear whether you checked these comments manually to guarantee that they are not abusive. 2. Difficult or ambiguous cases are most likely not part of this set.

The dataset splits should have been done chronologically - not random. Language and topics evolve and a time-based splitting would be more realistic for simulating an in-use system.

**Documentation:**

Instead of spending half of the paper on results for a not really novel method on a binary classification task, the authors should have documented their dataset in more detail in the paper instead of supplemental material. Following [29] is a good idea, but the main points should be in the paper, not in supp. material. Especially limitations and ethical considerations for use, etc. need to be part of the paper.

**Ethics:**

Social media content always has the problem of not being really anonymous, since one can easily search for a comment to find the user who posted it. Therefore, a more in-depth discussion on the risks would have been appreciated.

**Relation To Prior Work:**

Authors mention that different formats/schemes could be problematic when joining multiple datasets.
There exists work, which does exactly this:

Julian Risch, Philipp Schmidt, and Ralf Krestel. 2021. Data Integration for Toxic Comment Classification: Making More Than 40 Datasets Easily Accessible in One Unified Format. In Proceedings of the 5th Workshop on Online Abuse and Harms (WOAH 2021), pages 157–163, Online. Association for Computational Linguistics.

In general, there is a lot of work on hate speech/toxicity/abuse/etc. datasets, especially in non-indic languages. Therefore, transfer learning approaches or approaches using machine translation should have been mentioned and it should have been demonstrated that they are significantly worse compared to a native indic annotated dataset.

**Summary And Contributions:**

The first half of the paper presents a collection of 150k social media posts in 5 indic languages annotated by experts based on their toxicity. The second half of the paper presents the results for binary classification using a pre-trained XLM-R model on abusive content detection.

---

> ### Author Response · Authors · 2022-08-13
> **Responses to Reviewer SYqh concerns and feedback**
>
> We thank reviewer SYqh for providing us with detailed feedback. We are happy to clarify any further concerns.
>
> **1. Sampling of non-abusive comments:** We wanted to clarify that non-abusive comments of MACD were not sampled randomly. We sourced all the comments in MACD from comments which were reported by users as offensive on the platform. We manually annotated all the 150K comments included with MACD. So, the non-abusive comments have also been annotated by at least two annotators and adjudicated by a third expert annotator wherever necessary. We will make this more clear in the updated version.
>
> **2. Ambiguous cases:** MACD has difficult and ambiguous cases because MACD contains comments which were reported as abusive by platform users (Line 114). Note that the annotation was done on these flagged comments to start with and therefore there ought to be many difficult and ambiguous cases which the annotators had to deal with.
>
> **3. Transfer learning and translation:** We performed the following experiment to evaluate the performance of models trained using non-Indic languages on MACD.
>
> We finetuned MuRIL model  (https://arxiv.org/pdf/2103.10730.pdf) which was pre-trained over 17 Indic languages over a combination of abusive datasets in English contributed by Davidson et al. (https://arxiv.org/pdf/1703.04009.pdf ), Founta et al (https://ojs.aaai.org/index.php/ICWSM/article/view/14991), and HateXplain (https://ojs.aaai.org/index.php/AAAI/article/view/17745).
>
> We combined the *hate* and *offensive* categories in these datasets for training a binary classification model.
>
> We then translated the evaluation set of MACD into English using google translate and evaluated the performance of this model on the English translated-MACD test sets. Following are the results (accuracy and F1) of this study:
>
> **Tamil (43.7 and 32.19), Malayalam (43.40 and 32.49), Kannada (45.13 and 33.04) and Hindi (45.31 and 32.57)**
>
> These results show that it is challenging to achieve competitive performance by using transfer learning (non-Indic to Indic languages) and translation based approaches. A possible reason is that abusive behaviour is driven by cultural, political and religious beliefs due to which models trained on non-Indic languages and context do not transfer so well for Indic context.
> Thus, we believe that large scale datasets like MACD can be instrumental in encouraging research in abuse detection for severely resource-impoverished Indic languages.
>
> Transfer learning and translation based methods can further help in improving the performance of these models and scaling them for more Indic languages.
>
> **4. Binary vs Multiclass classification:** We apologise if we were not clear in the paper about the task. MACD is a binary classification dataset only with each comment annotated as “abusive” or “not-abusive”. Therefore, all our experiments are also binary classification experiments.
>
> **5. AbuseXLMR:** We agree that AbuseXLMR may not qualify as a novel technical contribution from modelling point of view. We will relax this claim in the next version of our paper. However, the idea of pretraining using 5 million Indic social media comments spreading over 15 Indic languages allows AbuseXLMR to improve upon the XLMR and MuRIL (Table 8) on multiple Indic datasets. To the best of our knowledge, such models do not exist for Indic languages. We sincerely hope that AbuseXLMR would become the standard model for further research into abuse detection in Indic languages.
>
> **6. Chronological splits:** This is a great suggestion. We are releasing chronological splits of MACD as well. (https://github.com/ShareChatAI/MACD/tree/main/dataset_chrono). Our baseline results (accuracy and F1 score) with the above chronological splits using AbuseXLMR are following:
>
> **Hindi - 87.43 and 86.78, Tamil - 87.47 and 86.99, Telugu - 90.42 and 90.38, Kannada - 87.16 and 87.16 and Malayalam - 83.17 and 81.82**
>
> We will conduct the experiments with other models also and include them in the next version of the draft.
>
> **7. Relation to prior work:**  Different Indic datasets follow different annotation guidelines for annotating abusive content and are often sourced from different platforms. This might result in inconsistencies in the annotations and affect the learning of the models. MACD circumvents this problem by having a large-scale dataset curated from a single source and annotated based on a consistent set of guidelines.
>
> **8. Ethics and Limitations:** We will expand upon the ethics, risks and limitations sections and move them from the supplementary section to the main paper to enhance clarity.
>
> We will also update the references with peer-reviewed versions.

---

> > ### Author Response · Authors · 2022-08-18
> > **Have we been able to address your concerns?**
> >
> > Dear reviewer SYqh,
> >
> > We would be grateful if you could let us know whether we succeeded in addressing your concerns and improving the overall understanding of our paper. If yes, we would appreciate if you could increase your rating. However, if you still have concerns, please guide us as to how we can improve upon them.
> >
> > Thank you!

---

### Official Review · Reviewer_VFRN · 2022-07-27
**This paper releases a large-scale and multilingual Indic language dataset**

**Rating:** 7
**Confidence:** 3
**Correctness:** The claims in the paper are generally…
**Clarity:** This paper is overall clearly written

**Strengths:**

- One of the largest abusive speech datasets for Indic languages, filling in the gap in the community of abusive comment detection.
- The AbuseXLR is a powerful social media dataset.
- The experiments are comprehensive and valid.
- Error analysis is interesting and has interesting insights


**Weaknesses:**

In the joint training experiment, the authors claim “the joint training (joint) improves the performance for all the languages over the monolingual (mono) training”. However, XLM-R (joint) does not outperform XLM-R (mono) in Hindi (Acc and F1) and Telugu (Acc). This should be justified.

**Additional Feedback:**

Please see strengths and weaknesses

**Documentation:**

The documentation is overall clear and easy-to-follow

**Ethics:**

I do not see outstanding ethic concerns

**Relation To Prior Work:**

Yes, the differences are clearly stated and justified

**Summary And Contributions:**

This paper releases a large-scale and multilingual Indic language dataset called Multilingual Abusive Comment Detection dataset (MACD) curated from a popular social media for the research community of abusive comment detection. Compared to the existing dataset, MACD is large-scale (150k textual comments) and multilingual (5 Indic languages). MACD is one of the largest abusive speech datasets for Indic languages. Further, the authors propose AbuseXLR for abusive comment detection. The experiments show the effectiveness of the AbuseXLR.

---

> ### Author Response · Authors · 2022-08-12
> **Responses to Reviewer VFRN concerns and feedback**
>
>  We thank reviewer VFRN for providing us with detailed feedback and are really glad that VFRN found our paper comprehensive and interesting.
>
> Joint training using mBERT improved on both Acc and F1 for all the languages over monolingual training (Table 6) while XLMR did not show improvement for Hindi language. A possible reason could be that since mBERT has better shared and aligned representations for various Indic languages due to which the differences due to language family (Dravidian vs Indo-Aryan) are less pronounced and thus all the languages help each other. We see similar phenomenon in zero-shot experiments (Column 2 and 3 in Table 4), where mBERT shows better zero-shot performance (Accuracy and F1) than XLMR when trained using Dravidian languages as *source* and evaluated on Hindi language as *target*.  A deeper and more comprehensive study is required to conclusively explain this phenomenon.

---

> > ### Author Response · Authors · 2022-08-17
> > **Thank you for the feedback**
> >
> > We thank reviewer VFRN for your valuable feedbacks !
> >
> > We will incorporate the above explanations in the next version of our paper.
> >
> > If we succeeded in addressing your concerns and improving the overall understanding of our paper, we would appreciate if you could increase your rating.
> >
> > Thank you !

---

### Official Review · Reviewer_erM3 · 2022-07-27
**Interesting dataset for Abuse Binary Classification task for 5 Indic languages**

**Rating:** 8
**Confidence:** 5

**Strengths:**

Interesting paper with thorough analysis across various axes.

Some strengths:
- Significant amount of data open-sourced for 5 important Indic languages for the abuse detection task
- Releases a good model (Abuse-XLMR) pretrained on large amount of real social media
- Interesting analyses across different dimensions like few-shot, zero-shot, cross-dataset, etc.

The paper was overall very well-written and easy to follow. Enjoyed reading it in detail.

**Weaknesses:**

I see the following couple of major weakness:

- Literature Survey for similar datasets is not well done. There seem to be atleast double the existing datasets mentioned in this paper.
- Choice of XLMR over other pretrained models is not very clear (could probably be addressed with more explanation)

There are a few other questions / comments, which I have put in the "Clarity" section.

**Additional Feedback:**

Very good paper overall, and valuable contribution to the Indic community ! ~But I have rated it 6 due to quite a many questions for which the answer is unclear/unknown.
I would be glad to increase my rating significantly if all (or atleast most) points mentioned in the review are addressed / explained in the paper.~

**Clarity:**

Overall, the paper was clear enough. I have put some of my questions, concerns, comments as well as things to clarify below:

1. L48: What is "data source limit"? (and how is it related to not enabling modeling by combining all open datasets?)

2. L56: For the sake on non-Indian audience, would be helpful to explain in detail why is there is only 1 language from the Indo-Aryan family and 4 from Dravidian, when Indo-Aryan is the second-most spoken-language family in the world (after European).

3. Fig. 2.d. shows 6 categories of abuse. Will that also be released? If not, please clearly mention that the dataset will only have pairs of comments and 0/1 to denote abuse/safe.

4. L173: Why not 80:10:10 ? (especially since we are working with low resource languages where more training is essential). 10% of 150k is 15k, which is already quite a significant number for test set (around 3k per lang). Would be helpful to explain why 60:20:20 was preferred or if possible, release an alternate standard split with 80:10:10. Although it is understandable that many researchers just use both training & val sets for training for practical purposes, still I believe 80:10:10 would be helpful.

5. L178: Why not AbuseMuRIL or Abuse-mBERT or AbuseIndicBERT ? Please mention what was the decision behind the choice of XLM-R to continue pretraining. Although I see in table 3 that XLM-R performs better than MuRIL in 3 langs, in the other 2 langs MuRIL is better, but footnote 6 says footnote requires paired corpora (hence difficult to pretrain). If this is correct, please explicitly mention so.

6. L183 (footnote 5): Since it was already mentioned that assuming language based on user profile is not reliable, why not use a language identification model or atleast determine the language based on the dominant unicode character ranges used whenever applicable? (like it was mentioned earlier). For example, for langs like Hindi, Nepali, Marathi, etc. since they all use Devanagari, a character set based match will not imply the language. But in other langs with dedicated scripts (like Dravidian languages, Gujarati, Panjabi, Odia, etc.), this could work very good. Would be great if you can share why this was not considered.

7. L198 (ref to table 3): L102 mentions a model called IndicBERT. Why was that not evaluated?

8. L207: Interesting study! Some language-wise interpretability analysis could make it more interesting. Like for example, why are different source languages helpful for different target languages in different models? (That is, for example, why does Kannada provide different best results in the 3 different models?)

9. L226: Do you mean you combine the train and val sets, and use that for training the joint model? If so, why?

10. L234: Since the continued pretraining was multilingual, why not also perform a multilingual fine-tuning experiment to see if it is better than the __joint__ results? Basically, why not try pr-mono+ft-joint?

11. Appendix Table 10: Interesting to see that there is not even 4% code-mixing. Was the comments in the dataset picked somehow such that this is kept as low as possible? Because predominantly, South Asian people on social media use roman text to write comments in their native language. Would be helpful to provide more info about this.


**Correctness:**

Mostly, all the experiments are well thought-through and does a pretty good job of benchmarking. (although there are some concerns mentioned in the below section)

One minor comment:
Please follow proper ISO 639-1 language code wherever applicable. This is the proper international standard.
For example, Kannada is "kn", not "Ka".

**Documentation:**

Went through the datasheet as well as glanced through the dataset CSV, looks good to me.

**Ethics:**

I do not see any ethical concerns per se. The authors seem to have ensured not to release any PII, as well as taken measure to mask sensitive info in the data.

**Relation To Prior Work:**

When searching for related abuse detection datasets for Indic langs, came across this previous work by the same organization:
https://www.kaggle.com/competitions/iiitd-abuse-detection-challenge/overview

Why is this not mentioned anywhere? How is this work different from that?
Does it have any overlap? Please explain in detail, this is quite essential.
Especially because it seems to have more than double the no. of languages covered in this work.

---

Also, came across another dataset which covers 3 of the 5 Indic langs evaluated in this work:
https://arxiv.org/abs/2106.09460

Why is this not mentioned at all? It seems to be of significant size (60k samples).
Please cover this as well.

**Summary And Contributions:**

The work makes the following contributions:

- A new large dataset in 5 Indic languages for abusive comment classification
- A new model called AbuseXLMR pretrained (continually on XMLR) on 15 Indic languages on private dataset
- Extensive benchmarking and baseline (incl. SOTA) results for the new MACD dataset

---

> ### Author Response · Authors · 2022-08-12
> **Responses to Reviewer erM3 concerns and feedback - Part 1**
>
> We thank reviewer erM3 for providing us with detailed feedback and glad that the reviewer found our work interesting and valuable. If our responses addressed your concerns, kindly consider increasing the ratings.
>
> **ISO 639-1:** We will update the draft with ISO 639-1 language code in the next version.
>
> **1. Data source limit:** We wanted to point out that in many earlier studies, data from multiple platforms like Twitter, Gab, Reddit, Facebook etc. have been combined. Further labels have been normalised across datasets. Such processes can introduce inconsistency and thereby affect the results. MACD circumvents this problem by having a large-scale dataset curated from *a single source* and annotated based on a *consistent set of guidelines.*
>
> **2. Languages:** We selected these five languages as these are the most dominant ones on our platform. In the forthcoming releases we plan to have another 5-7 languages.
>
> **3. Abuse categories:** We are releasing only binary labels. We will make this more clear in the next version of the draft.
>
> **4. Dataset splits:** We have released 80:10:10 splits for MACD on our official github repo (https://github.com/ShareChatAI/MACD/tree/main/dataset_80_10_10). We ran experiments on this split using AbuseXLMR and observed the following accuracy and F1.
>
> *Hindi - 88.53 and 88.52, Tamil - 88.83 and 88.81, Telugu - 91.63 and 91.60, Kannada - 88.00 and 88.00, Malayalam - 88.23 and 88.14*. We shall attempt  to conduct the experiments with other models also and include them in the next version of the draft.
>
> **5. XLMR vs others:** We selected XLMR because XLMR performed well (Table3) and does not require consecutive sentences corpora as it does not use next sentence prediction task (NSP). MACD comments are generally single-sentence comments and consecutive comments are not coherent and thus it was not possible to prepare a dataset for the NSP task. MuRIL and mBERT require consecutive sentences as they use NSP task. We will make this more explicit in the next version of the draft.
>
> **6. Language Identification:** We did use the character set to identify the languages (Line 122). However, as pointed out by the reviewer, this may not hold for languages which share a similar character set. Thus, we double-checked the MACD dataset language with human annotation also.
>
> **7. IndicBERT:** IndicBERT is based on ALBERT (https://arxiv.org/pdf/1909.11942.pdf) which is a compressed version of BERT with reduced parameters. Thus, we did not consider it for this study as we wanted to compare models with similar capacity. However, we will add the results in the next version of our draft.
>
> **8. Language wise analysis:** The transferability of *abuse detection* models trained on different languages in zero/few shot settings is an interesting area and requires deeper analysis. We will consider this for future work.
>
> **9. Splits combination:** We combined the training sets of all languages to form a new training set. Same process was followed for the validation set. We did not combine the train and val set with each other. We will make this more clear in the next version of our draft.
>
> **10. Pr-mono+ft-joint:** This is a great suggestion. We will conduct these experiments and update them in the next version of our draft.
>
> **11. Code-Mixing:** We intentionally removed those comments which were code-mixed in nature since we wanted MACD to enable research and analysis for Indic languages. In future we plan to release an independent dataset containing code-mixed languages for studying the interplay between languages.

---

> > ### Author Response · Authors · 2022-08-12
> > **Responses to Reviewer erM3 concerns and feedback - Part 2**
> >
> > **Relation to prior work:** Dataset released as part of Kaggle challenge was not peer-reviewed so we did not mention it. However, for completeness, we will refer to it in the next version of our paper. MACD does not contain overlap with that dataset and is different in following ways:
> >
> > *Consistent annotations:* Annotated by 2 annotators with disagreements being resolved by a third annotator. Kaggle dataset was annotated by only one person.
> >
> > *Language annotations:* MACD contains human-annotations for language of the comment along with using unicode based language identification. This ensures reliable language-specific and cross-lingual study. Kaggle dataset uses the language of the user which is not reliable in nature.
> >
> > *Code-mixing:* MACD has been designed for Indic language study with a small percentage of code-mixing. However, the Kaggle dataset does not take this aspect into account.
> >
> > *Rich social graph:* MACD is much richer as it also contains the mapping to the *user* and original *post* and thus allows for modelling relationships.
> >
> > *Publicly available:* MACD is publicly available for research purposes but Kaggle dataset is not available now for further research.
> >
> > The dataset mentioned in this paper (https://arxiv.org/abs/2106.09460) is code-mixed in nature. We compared MACD with datasets which are not code-mixed in nature in the main paper because our objective was to create datasets specific to Indic languages. In Appendix Table 11, we compared some code-mixed datasets and we will include this one also in the final version of the paper.

---

> > > ### Comment · Reviewer_erM3 · 2022-08-13
> > > **Thanks for the clarifications.**
> > >
> > > I have no more concerns about relation to prior works.

---

> > > > ### Author Response · Authors · 2022-08-13
> > > > **Addressing remaining two concerns.**
> > > >
> > > > We want to thank reviewer erM3 for their time to provide us additional feedbacks and clarifications.
> > > >
> > > > **Point-6 (Part-1):**
> > > >
> > > > Sincere apologies for misunderstanding your question. We agree that we could have used language-identification in Section 4 also. However, since in Section 4, we are pretraining the XLMR model for overall Indic social media domain, the accurate determination of the language of comments did not effect our training process as all the data was merged together before pretraining. Thus, we did not repeat the same process of language identification and considered the *user-profile-language* as coarse-level ground-truth to ensure that pretraining corpus is linguistically diverse for enabling the use of AbuseXLMR across different Indic languages.
> > > >
> > > > **Point-11 (Part-1):**
> > > >
> > > > We agree with the reviewer that Indic languages expressed in English alphabets is relevant and practical field of research. We will consider this suggestion in the future releases of MACD. We also believe that transliterating the MACD dataset into English alphabets could also act as bootstrap data for this research problem. However, a deeper study is required to establish the effectiveness of this approach.

---

> > > > > ### Author Response · Authors · 2022-08-18
> > > > > **Have we been able to address your second round of concerns?**
> > > > >
> > > > > Dear reviewer erM3,
> > > > >
> > > > > We would be grateful if you could let us know whether we succeeded in addressing your second round of concerns and improving the overall understanding of our paper. If yes, we would appreciate if you could increase your rating. However, if you still have concerns, please guide us as to how we can improve upon them.
> > > > >
> > > > > Thank you!

---

> > > > > > ### Comment · Reviewer_erM3 · 2022-08-22
> > > > > > **Regarding remaining concerns**
> > > > > >
> > > > > > Thanks for the responses. It would be great if you can incorporate the suggestions to which we have agreed upon, and upload a draft version (before the discussion phase closes) so that all the reviewers have a look at the revised version.
> > > > > >
> > > > > > Few more comments:
> > > > > >
> > > > > > > We agree with the reviewer that Indic languages expressed in English alphabets is relevant and practical field of research. We will consider this suggestion in the future releases of MACD.
> > > > > >
> > > > > > Thanks. It would be great to include your (now deleted) Kaggle dataset as well in your future works, as it seems to have significant amount of roman data. The problem gets more challenging with real-world roman data, as the style using which one writes their Indic language using English alphabet is quite personal and would have significant amount of variations in spelling, which is more practical to solve (like you have rightly mentioned).
> > > > > >
> > > > > > Please include this in the limitations (or future works) subsection.
> > > > > >
> > > > > > > We agree that we could have used language-identification in Section 4 also. However, since in Section 4, we are pretraining the XLMR model for overall Indic social media domain, the accurate determination of the language of comments did not effect our training process as all the data was merged together before pretraining. Thus, we did not repeat the same process of language identification and considered the user-profile-language as coarse-level ground-truth to ensure that pretraining corpus is linguistically diverse for enabling the use of AbuseXLMR across different Indic languages.
> > > > > >
> > > > > > Got it. Please mention this in the revised version.
> > > > > >
> > > > > > > Abuse categories: We are releasing only binary labels.
> > > > > >
> > > > > > As other reviewers have also pointed out, it would be commendable if you can also release the abuse classes as well (the labels behind Fig.2.d). This would make the contribution of this work stronger. Please note that the abuse classes have much more practical use.

---

> > > > > > > ### Author Response · Authors · 2022-08-25
> > > > > > > **Revised draft with suggested changes**
> > > > > > >
> > > > > > > Dear reviewer erM3,
> > > > > > >
> > > > > > > We have incorporated the suggested changes and have revised the draft. The changes are marked in "blue" (will remove them in the camera-ready version).
> > > > > > >
> > > > > > > Kindly have a look. We would be glad to provide further clarifications.
> > > > > > >
> > > > > > > We thank you for your valuable feedback and help in improving the draft.

---

> > > > > > > > ### Author Response · Authors · 2022-08-27
> > > > > > > > **Revised manuscript submitted.**
> > > > > > > >
> > > > > > > > Dear Reviewer erM3,
> > > > > > > >
> > > > > > > > As per your last suggestion, we have tried to address the comments from the referees and accordingly revised the manuscript (all changes have been marked in blue). We would be grateful if you could indicate whether this effort qualifies for an improved rating. We would be also happy to address any pending final issue.
> > > > > > > >
> > > > > > > > Thank you.

---

> > > > > > > > > ### Comment · Reviewer_erM3 · 2022-08-27
> > > > > > > > > **Thanks, most concerns are addressed**
> > > > > > > > >
> > > > > > > > > Thank you. I have no more questions as of now. Most of them are clarified.
> > > > > > > > >
> > > > > > > > > I still think the following couple of minor changes are not yet incorporated:
> > > > > > > > >
> > > > > > > > > ```
> > > > > > > > > 7. IndicBERT: IndicBERT is based on ALBERT (https://arxiv.org/pdf/1909.11942.pdf) which is a compressed version of BERT with reduced parameters. Thus, we did not consider it for this study as we wanted to compare models with similar capacity. However, we will add the results in the next version of our draft.
> > > > > > > > > 10. Pr-mono+ft-joint: This is a great suggestion. We will conduct these experiments and update them in the next version of our draft.
> > > > > > > > > ```
> > > > > > > > >
> > > > > > > > > Please do if possible. (optional though)
> > > > > > > > >
> > > > > > > > > But overall, I think this work makes a good contribution towards the advancement in NLP for low-resource languages.
> > > > > > > > > I also believe that the future works by this team has the potential to make much more significant contributions to the Indic-community.
> > > > > > > > > I have altered my rating of the paper as per my current judgement about the immediate potential impact possible by the work.
> > > > > > > > >
> > > > > > > > > In conclusion, few general points, addressing which would have made the contributions stronger:
> > > > > > > > >
> > > > > > > > > 1. As pointed out by other reviewers, the authors could have dedicated (manual) time in constructing a reasonably challenging test split. Having 85-90% accuracies already makes one think that the problem is already close to being reasonably solved. More broadly, the result that a 550M param model (XLM-R) after going through a large-scale pretraining & fine-tuning performs only 5% better (on avg) than a TF-IDF model suggests something serious. Wondering if the TF-IDF model was more carefully studied, if it could have reduced the gap even more.
> > > > > > > > > 2. Since the annotations were manual, it would have been more practical if the dataset involved significant amount of roman and code-mixed data as well, instead of just filtering them off. In such real-world cases, I am guessing that models like MuRIL would have performed significantly better. Although I can understand that their current research focuses only on comments which are written (almost) purely in Indic scripts, my point here is that it may only cover a fraction of the online South Asian audience's behavior.
> > > > > > > > > 3. No information was provided regarding the nature of the pretraining dataset. What % of code-mixed or romanized content does it have (if any)? What were the list of all 15+ languages on which AbuseXLMR was pretrained? What was the amount of data per language? What % of the pretraining data covers the 5 languages on which the model is fine-tuned? Also, the fact that this is a private dataset (which cannot be reconstructed/collected by anyone) makes this research non-reproducible. This should ideally be acknowledged in the limitations.

---

> > ### Comment · Reviewer_erM3 · 2022-08-13
> > **Thank you for the answers. Couple of clarifications:**
> >
> > Regarding point-6 in part-1:
> >
> > This comment I had mentioned was for section-4 (processing of pretraining data for 15 Indic langs), not for section-3 (the fine-tuning data for 4 Indic langs). Please check once again.
> >
> > Regarding point-11 in part-1:
> >
> > Probably I did not put it clearly. By "code-mixing", I meant mixing of roman and Indic scripts (not English and Indic langs). On social media, South Asians predominantly write using roman script (as compared to native scripts). Hence, although the alphabet is that of English, the language would still be Indic. So my point was that it would have been much more valuable if such roman data were not discarded, as it makes the work more practical.

---

### Official Review · Reviewer_ctsU · 2022-07-28
**MACD: Multilingual Abusive Comment Detection at Scale for Indic Languages**

**Rating:** 6
**Confidence:** 3
**Clarity:** This paper is well-written, clearly s…

**Strengths:**

1. This paper provides a large-scale and well-annotated multilingual dataset.
2. The proposed AbuseXLMR performance is better against baselines.
3. This paper discusses all-side experiment settings.

**Weaknesses:**

1. The relationship between diversity and user distribution is not clear.
2. Age distribution of annotators is concentrated in young people while abusive language is likely to be different between different age groups.
3. The AbuseXLMR is pretrained with social media data while it is pretrained with a larger scale of data. I hope the authors may discuss the effects of the scale of pretrained data and different domains of pretrained data.
4. In the error analysis section, is the XLM-R pretrained with the social media data. If not, why not discuss the proposed AbuseXLMR.

**Additional Feedback:**

None.

**Correctness:**

The submission is highly likely to be constructed in a sound way. As experts are claimed to provide the annotation and the analysis of dataset is comprehensive and convincible.

**Documentation:**

This paper provides a general insight of data collection. And the the organization of the dataset is sufficient and is available to access.

**Ethics:**

No ethical concerns.

**Relation To Prior Work:**

This paper discusses previous work well and provides statistical data of previous work.

**Summary And Contributions:**

This paper introduces a competitive multilingual dataset for abusive comment detection for Indic language and an effective model for solving multilingual abusive detection. The dataset has a significantly larger scale than previous datasets. The annotation of the dataset is convincible as experts are claimed to be involved in the work and the measurement of annotator agreement proves the consistency of annotation. The proposed approach has novel parts-pretrained over social media data. The authors conduct extensive experiments over the MACD dataset and evaluate the AbuseXLMR model.

---

> ### Author Response · Authors · 2022-08-12
> **Responses to Reviewer ctsU concerns and feedback**
>
> We thank reviewer ctsU for providing us with detailed feedback. We are happy to clarify any further concerns.
>
> 1. **Relationship between diversity and user distribution:** We believe that by collecting MACD dataset from more than 70K social media users, we have sampled a wide range of writing nuances like spelling mistakes, grammatical mistakes, use of abbreviations and emojis along with various social sensitivities and beliefs (religious and political views) which make MACD highly diverse.
>
> 2. **Age distribution of annotators:** The annotation team for MACD belongs to professional and experienced annotators employed by ShareChat on contractual basis and are trained with strict platform guidelines to mitigate personal bias in the annotation. Moreover, two annotators annotated every sample and a third (more experienced) annotator helped in resolving the disagreements.
> Additionally, as reported here (https://www.similarweb.com/website/sharechat.com/#geography), majority of the traffic on ShareChat belongs to the age group of 18-24 years. Thus, an average age of 25 years (Line 134) for our annotation team aligns well with ShareChat users. We believe these measures encourage unbiased and accurate annotations.
>
> 3. **Pretraining and Error Cases:** Superior performance of AbuseXLMR over XLMR highlights the importance of pretraining with data sourced from similar domains (Indic social media data). Similar observations are also made by HateBERT (https://arxiv.org/abs/2010.12472) for the English language. These observations highlight that domain-specific pretraining followed by finetuning helps in improving the performance of these models. We will further elaborate on this point in the next version of our paper. We will also discuss the error cases of AbuseXLMR in the updated version of our paper.

---

> > ### Comment · Reviewer_ctsU · 2022-08-13
> > **Thank you for the response.**
> >
> > I have no more concerns about the *Relationship between diversity and user distribution*, *Error Cases*, and *Age distribution of annotators* parts.Thank you for the additional analysis, it is convincing and please add it to the paper.

---

> > > ### Author Response · Authors · 2022-08-13
> > > **Thank you for the feedback.**
> > >
> > > Thank you ctsU for your valuable feedback and prompt response !
> > >
> > > We will add the above analysis in the next version of our paper. If we succeeded in addressing your concerns and improving the overall understanding of our paper, we would appreciate if you could increase your rating.
> > >
> > > Thank you !

---

### Official Review · Reviewer_sUDJ · 2022-07-29
**Abusive language dataset with rather limited utility**

**Rating:** 5
**Confidence:** 5
**Clarity:** The paper is very well-written and ea…

**Strengths:**

The paper has the following strengths:
1. It tackles the very relevant problem of abuse detection on social media, which is of consequence to a lot of stakeholders, e.g., regulators, users, social media companies, etc.
2. The dataset released is of a good size with focus on Indic languages, which traditionally don’t have a lot of good datasets available.
3. The dataset collection and sampling strategy is robust in terms of ensuring that some users don’t dominate the dataset and that the dataset is not simply composed based on keywords. Other datasets in the field of abuse detection have been marred by such issues.
4. The experiments conducted are of a good breadth with few shot and zero shot setups taken into account. Results are reported in a systematic manner.

**Weaknesses:**

The paper has the following weaknesses:
1. The main weakness of the paper is that it really doesn’t build upon recent research in the field, along multiple facets in fact. The annotation guidelines are such that they really encourage non-binary labeling; there is a significant amount of subjectivity in what is considered abuse when it comes to beliefs, practices, and religion. Recent research (https://arxiv.org/pdf/2106.05664.pdf) has used best-worst scaling to account for such subjectivity in annotations; the authors have neither cited the paper nor taken learnings from it in creating their dataset.
2. Furthermore, while the authors create a good size dataset from a social media platform, they could have gone further to make the dataset richer in terms of also providing the social graph information of users; there is no discussion of this even though state of the art abusive language detection methods (https://arxiv.org/pdf/1904.04073.pdf) hugely benefit from social graph information, specially when these datasets come from social media platforms.
3. The AbuseXLMR model that the authors release already achieves a ~90% performance on the dataset. Keeping the inter-annotator agreement in view, I wonder what is left to be achieved on the benchmark anyway? The benchmark doesn’t offer a significant challenge. Additionally, the authors mention that misspellings and obfuscation is a problem for the models but they don’t bring into the discussion any past work on character-level composition to tackle the problem (https://arxiv.org/pdf/1809.00378.pdf).
4. ~Authors are releasing a balanced dataset, which they claim is a desirable trait as per them. I do not necessarily see the proper reasoning for this; abusive language is rather sparse in natural settings, so the dataset is far from what would be considered a natural distribution. Why is balancing a good thing here? Some more motivation around it would have been good to have.~ This point was addressed, authors provided results on splits with near-natural distribution.

**Additional Feedback:**

None

**Correctness:**

I found the dataset sampling strategy to be sound and the experiments to be well-conducted.

**Documentation:**

I found the documentation to be sufficient.

**Ethics:**

The authors have taken steps to anonymize the dataset. My only concern is how is deletion of comments handled?

**Relation To Prior Work:**

See weaknesses section; there is quite a lot of work that authors could have built upon to enhance the quality of the dataset they are releasing.

**Summary And Contributions:**

The paper tackles the problem of abusive language detection in social media with focus on Indic languages, which are traditionally not resource rich in terms of datasets. To that effect, the paper makes the following contributions:
1. Creates an annotated dataset of ~150k comments taken from an Indic social media platform, i.e., ShareChat
2. Presents experiments with several state of the art multilingual models to demonstrate the efficacy of their domain adapted version of XLM-R, i.e., AbuseXLMR.

---

> ### Author Response · Authors · 2022-08-12
> **Responses to Reviewer sUDJ concerns and feedback**
>
>
> We thank reviewer sUDJ for providing us with detailed feedback. We are happy to clarify any further concerns.
>
> 1. **Annotation Guidelines:** We agree with reviewer that beliefs, practice and religion can introduce subjectivity and bias in annotations. We tried to mitigate this bias by following methods:
>
>     1.1 Multiple Annotators: Every sample is annotated by 2 annotators and a third expert annotator (more experienced) resolves the conflicts in case of disagreements.
>
>     1.2 Native Annotators: The 3 annotators are native speakers of the language and thus they are comfortable with the language and overall social context of the region.
>
>     1.3 Professional Annotators following platform guidelines: The annotators recruited for this study are professionals who have been employed by ShareChat for moderating social media comments. They have vast experience in moderating social media comments in a consistent manner following the platform guidelines.
>
>     1.4 Ruddit Paper: - Ruddit (https://arxiv.org/pdf/2106.05664.pdf) provides great insights. We will refer to this work in the updated version of our draft and consider this as future work to further improve the annotations of MACD.
>
> 2. **Social Graph:** We have released the masked identifiers for users who made the comment and masked identifiers of the original post on which the comment was made to further enrich the MACD dataset with social graph in our official repository (https://github.com/ShareChatAI/MACD/tree/main/dataset_meta). Various social graphs can be constructed from these identifiers including user-post, user-user and post-post. While our main objective here was to release the dataset and present its basic potential to detect abuse in Indic languages, future studies (including ours) can use these social graphs to further enrich the model.
> 3. **MACD Challenges:** We report the failure cases in Section 6 and appendix to highlight the instances where leveraging external knowledge graphs, social graphs and improved modelling could be helpful. We hope that future works explore these interesting directions using our MACD dataset.
> 4. **Balanced Dataset Experiments:** We agree that abusive language is pretty sparse and a balanced dataset does not represent natural settings. However, we respectfully argue that we chose to release a balanced dataset as balanced datasets are considered to facilitate better learning. To empirically verify this, we resampled the training set of MACD for Hindi language into two new subsampled sets -
>
>     a) *Balanced train set* - Equal ratio of abusive and non-abusive samples (1:1)
>
>     b) *Unbalanced train set* - Skewed ratio between abusive and non-abusive samples (1:5)
>
>     Both the train sets contain the same number of samples for fair comparison. For validation and test set, we used the original MACD splits.
>
>    We note that AbuseXLMR trained using *balanced train set* performs **~2.7% better in both F1 and Accuracy** over the same model trained using *unbalanced train set*. This shows that balanced train set helps in better learning.
>    Using techniques like upsampling, data-augmentation can help in reducing this gap and we hope that researchers interested in similar methods could subsample MACD to study imbalanced scenarios. A recent CSCW paper (https://arxiv.org/pdf/1805.08168.pdf) also presents further evidence for choosing a balanced dataset.
>
>      A balanced dataset also allows MACD to capture comprehensive types and a variety of abusive comments. Large scale MACD with.
>      skewness toward non-abusive comments will severely defeat the purpose of the large scale.
>
>      We will also release an unbalanced version of MACD splits dataset for facilitating study on unbalanced dataset.
>
> 5. **Comments Deletion**: As noted by the reviewer, we have taken strict measures to anonymize the dataset for any personally identifiable information (PII). Additionally, we would provide an opt-out form where users could request for an explicit deletion of comments from the MACD dataset.

---

> > ### Comment · Reviewer_sUDJ · 2022-08-13
> > **Thank you for the response.**
> >
> > 1. **Regarding annotation guidelines**: While I understand that authors have taken measures to reduce subjectivity, it doesn't fundamentally address the issue raised that binary labels do not capture the complexity of abuse detection well.
> >
> > 2. **MACD challenges**: I do understand that there are failure cases reported in the paper, some of them could be addressed by character level composition as suggested, which hasn't been talked about in the paper even though it is existing work. Moreover, even though there are avenues to explore with KGs and social graph based approaches, still is there really scope given that we are already at ~90% performance, keeping in view the inter-annotator agreement.
> >
> > 3. **Balanced dataset experiments**: Training with balanced set does improve performance, but isn't that the fallacy here. In a real-world setting, any model we deploy will always be trained and predicting on a natural distribution where abuse is a small subset of all the comments. So, by training and evaluating on balanced datasets, we are actually not getting an accurate picture of how the system would perform in a natural setting, which reduces the real-world applicability of the paper.

---

> > > ### Author Response · Authors · 2022-08-18
> > > **Responses to the remaining concerns**
> > >
> > > **1. Regarding annotation guidelines:** We completely agree with the reviewer that binary labels do not capture the complexity and nuances of abusive behaviour.  However, considering the rampant spread of abusive behaviours on social media platforms and its harmful effects on the society, the immediate and urgent goal for social media platforms like ShareChat is to track and flag abusive posts *accurately and quickly* rather than delving into the nuanced characteristics of the abuse and its subtypes. One of the urgent needs for this is to aid the moderators of the platform and reduce their psycho-physical load in finding and flagging abusive posts manually from millions of posts received per day. Hence, the current binary design (though not comprehensive) has lot of value and could be an ideal starting point toward ensuring safe interactions on social media, particularly for Indic languages.
> > >
> > > However, we acknowledge that we have just scratched the surface with this binary design and annotating MACD with guidelines proposed in the Ruddit paper and fine-grained annotations would certainly increase the value of MACD tremendously and we would consider this as future work.
> > >
> > > **2. MACD challenges:** In our paper, we successfully examined and showed two methods for further improving the performance on MACD over existing state-of-the-art models like MuRIL and XLM-R:
> > > 1) *Pretraining:* In Table 1, we showed that pretraining of XLMR on social media dataset improved the F1 scores (nearly 1.5% absolute for majority of languages). Thus, pretraining is one approach which helps to improve the performance on MACD over previous models.
> > >
> > > 2) *Joint Training:* Similarly, in Table 6, we showed that joint training and pretraining over MACD also helps to improve the performance over monolingual training. Thus, leveraging cross-learning between languages also looks promising in furthering MACD performance.
> > >
> > > Both the above methods show that MACD performance is not saturated and there are opportunities for improving the performance across languages.
> > >
> > >
> > > 3) *Low data settings:* In Table 4 and Table 5, we also show that current models do not perform satisfactorily under zero-shot and few-shot settings. These results further position MACD as an important benchmark for further research in low-data settings, especially for Indic languages.
> > >
> > > Given the above points, we humbly argue that MACD still has a lot to offer in terms of research opportunities and considerable gains remain to be achieved.
> > >
> > > **3. Balanced dataset experiments:** We understand your concern and agree that balanced datasets, though useful for training better models, are not always available for developing abuse detection algorithms in real-world settings. To simulate these settings, we are releasing an unbalanced version of the MACD dataset (MACD-U) with a 1:5 ratio of abusive vs non-abusive samples for all the three splits (https://github.com/ShareChatAI/MACD/tree/main/dataset_unbalanced):
> > >
> > > Our baseline F1 macro scores on these split using AbuseXLMR are:
> > >
> > > *Hindi: 85.69, Tamil: 84.48, Telugu: 88.86, Kannada: 85.74, Malyalam: 82.81*
> > >
> > > *Since, the dataset is imbalanced, we report F1 macro score.*
> > >
> > > We observe a drop of nearly 3% F1 score in all the languages with these new splits. We hope that these splits would be instrumental in studying the real-world settings. We will explicitly include a section in the main paper for explaining these splits for clarity.

---

> > > > ### Author Response · Authors · 2022-08-22
> > > > **Second round answers**
> > > >
> > > > Dear reviewer sUDJ,
> > > >
> > > > We would be grateful if you could let us know whether we succeeded in addressing your second round of concerns and improving the overall understanding of our paper. If yes, we would appreciate if you could increase your rating. However, if you still have concerns, please guide us as to how we can improve upon them.
> > > >
> > > > Thank you!

---

> > > > > ### Comment · Reviewer_sUDJ · 2022-08-22
> > > > > **Thank you for your response.**
> > > > >
> > > > > **1. Regarding annotation guidelines:** I very much agree that automated abuse detection is the need of the hour. However, what I don't see is why binary labels are the best way to fulfill that need. In fact, if anything, I feel social media platforms require granular control over abusive language, and that comes from having systems trained to understand the spectrum of abusive language, from benign to borderline to explicitly abusive. That way, platforms can do better moderation at a granular level instead of over-enforcing or under-enforcing. I am not satisfied by the arguments that binary labels solve some need of the hour problem that otherwise won't be solved.
> > > > >
> > > > > **2. MACD challenges:** The point is not saturation. Indeed the performance on the dataset can be improved with more pre-training or more feature engineering that is tuned to the dataset. However, I don't necessarily see a *novel* challenge here than future work trying to do more of dataset-specific feature engineering in order to improve performance further. In other words, I would love to know from the authors, what are the general challenges that the dataset presents that will require some breakthrough? One problem that the authors highlighted is spelling variations, but as I already pointed out, there is existing work on it in the field of abuse detection. What other gaps do authors see that will require novel contributions? Even in the zero-shot and few-shot settings, we have already have very good machine translation systems that make the need for a lot of data in all languages redundant.
> > > > >
> > > > > **3. Balanced dataset experiments:** Not only are balanced datasets rarely available, in fact, no matter what dataset you use in real-world setting, it will always reflect a natural distribution. This is because language keeps evolving over time and whatever training samples you have for abusive class will always represent a small proportion of the space of comments you could be running your model on. So, it is generally better to report performance on near natural distribution to actually give a better idea of how the model will perform in a real-world setting. Thanks for now providing experiments regarding this, I have raised my score to 5.

---

### Meta-Review · Area_Chair_Eh6h · 2022-09-10

**Recommendation:** Accept
**Confidence:** 4

**Metareview:**

This paper creates MACD, a large-scale (150K), human-annotated, multilingual (5 languages) dataset for abusive language detection in social media. The dataset has a focus on Indic languages, which are low-resource for this task. The paper presents in detail the sampling and annotation process. In addition, the paper presents AbuseXLM-R, a domain adapted model that achieves state of the art. The paper is well written and contains detailed experimentation. Although the paper uses binary labeling, it includes several annotators, to attenuate subjective bias and disagreement. The size and diversity of the dataset make it a valuable contribution to the community.

*pros*

* the paper takes the challenging problem of online abuse in social media, a very relevant topic.
* the dataset size is large and focused on low-resource languages.
* the dataset has been sampled with care, to ensure its representativity
* the dataset is carefully anonymized to remove PII


*cons*
* the dataset is presented in a "balanced" distribution, which is not representative of the skewness and low-prevalence of real-life abusive content.
* the dataset focuses on binary labels whereas recent research suggests that continuous scales are more resilient to bias.

---

### Decision · Program_Chairs · 2022-09-16

Accept